# Heart Ferroportin Protein Content Is Regulated by Heart Iron Concentration and Systemic Hepcidin Expression

**DOI:** 10.3390/ijms23115899

**Published:** 2022-05-24

**Authors:** Betty Berezovsky, Jana Frýdlová, Iuliia Gurieva, Daniel W. Rogalsky, Martin Vokurka, Jan Krijt

**Affiliations:** 1Institute of Pathophysiology, First Faculty of Medicine, Charles University, 128 53 Prague, Czech Republic; bettyberezovsky@gmail.com (B.B.); jana.frydlova@lf1.cuni.cz (J.F.); gurieva.jul@seznam.cz (I.G.); martin.vokurka@lf1.cuni.cz (M.V.); 2Queen Elizabeth Hospital, Birmingham B15 2GW, UK; daniel.rogalsky@nhs.net

**Keywords:** iron metabolism, ferroportin, hepcidin, hemojuvelin, myocardium

## Abstract

The purpose of the study was to investigate the expression of ferroportin protein following treatments that affect systemic hepcidin. Administration of erythropoietin to C57BL/6J mice decreased systemic hepcidin expression; it also increased heart ferroportin protein content, determined by immunoblot in the membrane fraction, to approximately 200% of control values. This increase in heart ferroportin protein is very probably caused by a decrease in systemic hepcidin expression, in accordance with the classical regulation of ferroportin by hepcidin. However, the control of heart ferroportin protein by systemic hepcidin could apparently be overridden by changes in heart non-heme iron content since injection of ferric carboxymaltose to mice at 300 mg Fe/kg resulted in an increase in liver hepcidin expression, heart non-heme iron content, and also a threefold increase in heart ferroportin protein content. In a separate experiment, feeding an iron-deficient diet to young Wistar rats dramatically decreased liver hepcidin expression, while heart non-heme iron content and heart ferroportin protein content decreased to 50% of controls. It is, therefore, suggested that heart ferroportin protein is regulated primarily by the iron regulatory protein/iron-responsive element system and that the regulation of heart ferroportin by the hepcidin-ferroportin axis plays a secondary role.

## 1. Introduction

Iron is an indispensable micronutrient that, in stark contrast to all other biometals, lacks a regulated pathway for its excretion from the human body [1]. Therefore, increased absorption of iron from the gastrointestinal tract or increased parenteral loading with iron due to repeated transfusions invariably leads to iron overload. Diseases caused by increased absorption of iron are known as hereditary hemochromatoses. The most severe forms of hemochromatoses are juvenile hemochromatoses, characterized by rapid iron overload during the first decades of life. If untreated, juvenile hemochromatosis patients typically die of heart failure [2]. Similarly, patients requiring repeated transfusions, such as beta-thalassemia major, frequently die of heart failure unless treated with iron chelators [3].

Despite the fact that iron overload can have a dramatic detrimental effect on heart function, the regulation of iron metabolism in the heart is relatively poorly understood. In every cell in the body, iron content is determined by a balance between cellular iron uptake and cellular iron export [4]. In the case of cardiomyocytes, several proteins such as transferrin receptor [5], DMT1 [6], calcium channels [7], or zinc transporters [8] have been postulated to play a role in iron import, while iron export is mediated by the protein ferroportin [8]. Ferroportin (FPN) represents the only protein capable of exporting iron from the cell identified so far [9,10,11]. The importance of ferroportin for iron metabolism is evident from the pathophysiology of hereditary hemochromatosis—it is very well established that enterocyte ferroportin exports iron from the duodenal enterocyte into plasma, and that this rate-limiting step of iron absorption is tightly controlled by the iron regulatory hormone hepcidin [12,13]. The interaction of hepcidin with ferroportin represents the cornerstone of mammalian iron homeostasis [14], and many studies have demonstrated that decreased hepcidin levels result in increased enterocyte ferroportin content and, as a consequence, iron overload [12,13,15].

Intriguingly, there is only limited information regarding the regulation of heart ferroportin protein content, and there is contradictory information regarding the overall role of ferroportin in cardiomyocyte iron homeostasis. One report suggested that cardiomyocyte-specific deletion of ferroportin has no effect on heart function or heart iron homeostasis [16], while data from another laboratory showed that cardiomyocyte-specific deletion of ferroportin in mice results in fatal heart failure [17]. In addition, the following new concept has recently emerged regarding the regulation of cardiomyocyte ferroportin: Instead of being regulated by systemic hepcidin, as is the case for enterocyte ferroportin, cardiomyocyte ferroportin has been postulated to respond primarily to cardiomyocyte-produced hepcidin [18,19]. In this context, it has been reported that heart hepcidin expression in mice is relatively high, with heart hepcidin RNA content being only thirty times lower than hepatic hepcidin RNA content [18]. Since hepatic hepcidin expression in mice fed a standard rodent diet is substantial, i.e., approximately comparable to the expression of typical reference genes, such a relatively high level of hepcidin mRNA content would indeed represent a potentially plentiful source of biologically active hepcidin peptide. Intriguingly, the regulation of mouse cardiac ferroportin protein has not yet been examined in conditions that are known to affect systemic hepcidin expression.

Although iron overload undoubtedly has a strong detrimental effect on heart function in hemochromatosis or thalassemia, it has, during the past decades, been recognized that the functional capacity of the heart can also be negatively influenced by iron deficiency. In its 2021 guidelines, the European Society of Cardiology recommends considering the intravenous administration of ferric carboxymaltose to heart failure patients with iron deficiency [20]. At present, it is difficult to conclude whether the benefit of this treatment results merely from the effect of iron on red blood cell production, or whether administration of ferric carboxymaltose directly affects cardiac iron homeostasis. It is not known how parenteral iron treatment affects cardiac ferroportin expression. In addition, there is only limited information on the response of heart ferroportin protein to iron deficiency, although iron deficiency is reportedly observed in up to 80% of patients with acute heart failure [20].

Because ferroportin is one of the key proteins in cellular iron metabolism, its expression is regulated at multiple levels. Apart from the very well-known hepcidin-ferroportin axis [13], ferroportin is also regulated post-transcriptionally by intracellular iron content, through the interaction of iron-regulatory proteins (IRPs) with an iron-responsive element (IRE) in the 5-untranslated region of ferroportin mRNA [9,10,11,21]. Similarly, to the regulation of ferritin protein subunits, which also have an IRE element in the 5-untranslated region of their mRNAs, iron overload decreases the binding of the IRPs to the IRE and thus unblocks mRNA translation. In the case of ferritin, this regulation is extremely efficient, as evidenced by the dramatic increase in ferritin protein following iron overload in experimental animals [22]. Interestingly, it has recently been reported that the degree of translational IRP/IRE-dependent activation of protein synthesis by iron is approximately similar for both ferritin and ferroportin [23]. In addition to the sophisticated IRP/IRE system, iron has also been reported to regulate ferroportin expression transcriptionally [24]. It is not known which of the two posttranscriptional regulation systems—hepcidin/ferroportin interaction or IRP/IRE interaction—plays a dominant role in the final adjustment of cardiomyocyte ferroportin protein content. Interestingly, there is also only limited information on the hierarchy of these two regulatory systems in the liver and spleen.

The purpose of the present study was to examine the response of heart ferroportin protein to changes in systemic hepcidin in experimental animals and to compare heart ferroportin regulation with ferroportin regulation in the liver and spleen. The results suggest that, in contrast to ferroportin protein regulation in the duodenum, heart ferroportin protein content is determined primarily by heart iron concentration, whereas the regulation by hepcidin plays a secondary role.

## 2. Results

### 2.1. Erythropoietin Administration Increases Heart Ferroportin Protein Content in Mice

Administration of a high dose of erythropoietin (EPO) to mice for four days resulted in the expected decrease in liver hepcidin expression (Table 1). Liver iron content was slightly decreased. Heart and liver ferroportin protein content was increased by the treatment (Figure 1A). As reported previously [25], the sizes of the heart and spleen ferroportin proteins were slightly different, with the spleen protein migrating more slowly (approximately 62 kDa for heart FPN and 65 kDa for spleen FPN). In accordance with previously reported observations [25], splenic ferroportin protein content was not increased by EPO administration (Figure 1A); this apparent lack of effect of EPO on spleen ferroportin was probably associated with the marked splenomegaly induced by EPO treatment (159 ± 35 mg in EPO group vs. 90 ± 32 mg in controls). In contrast to the liver, administration of erythropoietin did not significantly decrease heart hepcidin (*Hamp*) mRNA content (Figure 1C). Heart *Hamp* mRNA content was by two to four orders of magnitude lower than liver *Hamp* mRNA content when using beta actin (*Actb*) as a reference gene; roughly similar results were obtained using *Gapdh* or *Rpl13a* reference genes (Appendix A).

### 2.2. Erythropoietin Administration Increases Heart Ferroportin Protein Content in Rats

In accordance with results obtained in mice (Figure 1), the effect of erythropoietin on heart FPN protein content was also observed in outbred Wistar rats. EPO increased FPN protein content in membrane fractions isolated from both the heart and liver; splenic FPN content was slightly decreased (Figure 2A). Similar to mice, EPO administration to rats resulted in marked splenomegaly (471 ± 81 mg for the control group and 1744 ± 211 mg for the EPO-treated group). While EPO administration dramatically decreased liver hepcidin mRNA content (Table 1), it did not significantly change heart *Hamp* mRNA content (Figure 2C). In contrast to inbred mice, heart *Hamp* mRNA content in outbred rats was more variable, relatively high, and, in some cases, comparable to *Fpn* mRNA content (Figure 2C).

### 2.3. Feeding of Low-Iron Diet Decreases Ferroportin Protein Content in Rat Heart, Liver, and Spleen

For the determination of the effect of iron deficiency on ferroportin protein content, we chose young Wistar rats placed for two months on an iron-deficient diet since this model reliably leads to profound sideropenia as well as to a decrease in liver hepcidin expression exceeding two orders of magnitude [26]. The treatment approximately halved the heart’s non-heme iron content, while liver and splenic iron content decreased to less than 10% and 5%, respectively (Table 1). Iron deficiency decreased heart ferroportin content (Figure 3A,B), as well as ferroportin protein content in the liver and spleen. In accordance with results reported for mice [25], the apparent kDa size of splenic FPN protein was slightly higher than the size of liver and heart FPN protein. Heart *Fpn* and *Hamp* mRNA was not significantly changed (Figure 3C). The results demonstrate that, in the heart as well as in the liver, the decrease in tissue iron content results in downregulation of ferroportin protein expression, despite the massive drop in liver *Hamp* mRNA content. The observed decrease in splenic ferroportin protein content could possibly be related to splenomegaly (731 ± 185 mg in the iron-deficient group vs. 472 ± 93 mg in controls) induced by the low iron diet as a response to iron-deficient erythropoiesis (hematocrit 43 ± 1% in the control group and 39 ± 1% in the iron-deficient group).

### 2.4. Heart Ferroportin Protein Content Is Increased in a Mouse Model of Juvenile Hemochromatosis

To investigate the expression of heart ferroportin in experimental juvenile hemochromatosis, we used mice with disrupted expression of hemojuvelin (*Hjv*−/− mice). These mice display decreased expression of hepcidin and marked liver iron overload. In accordance with previously published data [27], their heart non-heme iron content is markedly elevated (Table 1). As can be seen in Figure 4A,B, heart ferroportin protein content in these mice was significantly increased, which probably reflects both the increase in heart iron content as well as the decrease in systemic hepcidin expression. An increase was also observed in liver and spleen ferroportin protein content. Heart hepcidin mRNA content in *Hjv*−/− mice was not significantly affected, while the content of *Fpn* mRNA was increased (Figure 4C).

### 2.5. Iron Carboxymaltose Injection Increases Ferroportin Protein in Mouse Heart, Liver, and Spleen

To investigate the effect of parenteral iron overload on ferroportin protein expression, mice were injected intraperitoneally with Ferinject (Vifor, Paris, France) at a dose of 300 mg Fe/kg ten days before sacrifice. The treatment increased liver iron content by more than an order of magnitude; heart iron content increased approximately fivefold (Table 1). Heart ferroportin protein content increased approximately threefold (Figure 5A,B). In accordance with recently published data [28], we observed a marked increase in liver ferroportin protein content (Figure 5A) in iron-treated mice. Intriguingly, despite the expected increase in liver hepcidin expression (Table 1), splenic ferroportin protein content was also increased (Figure 5A). Since a recent study described a decrease in splenic ferroportin protein in mice placed on an iron-enriched diet [28], the current result apparently indicates that, in comparison with dietary iron overload, intraperitoneal injection of a high dose of iron carboxymaltose affects different cells or different regulatory pathways in the spleen. In contrast to *Hjv*−/− mice, which display a decrease in splenic non-heme iron content [27], the spleen iron content in Ferinject-injected mice increased more than ten-fold (Table 1), suggesting different handling of dietary iron and Ferinject-derived iron by spleen cells. As noted previously [25], ferroportin protein content in the spleen was markedly higher than ferroportin protein content in the liver, despite approximately similar *Fpn* mRNA content (Appendix A). Injection of Ferinject did not affect heart *Hamp* mRNA content but increased heart *Fpn* mRNA content (Figure 5C) and liver *Fpn* mRNA content (Appendix A).

## 3. Discussion

The main aim of the study was to examine the effect of changes in liver hepcidin expression on heart ferroportin protein content. In particular, it was of interest to determine the effect of liver hepcidin downregulation since it has recently been postulated that cardiomyocyte ferroportin might be regulated by cardiomyocyte-derived hepcidin rather than by systemic hepcidin.

From the physiological point of view, the most potent negative stimulus of liver hepcidin expression in mice is the increase in erythropoietic activity—administration of a high dose of erythropoietin to mice has been repeatedly demonstrated to decrease hepatic hepcidin expression by several orders of magnitude [29,30,31]. In the presented experiments, administration of erythropoietin, which was associated with a dramatic decrease in hepatic hepcidin expression, resulted in increased cardiac ferroportin content. Since cardiac hepcidin expression was not affected by erythropoietin treatment, this result strongly suggests that cardiac ferroportin is regulated by systemic hepcidin rather than by cardiomyocyte-derived hepcidin.

Although the regulation of cardiac ferroportin by locally synthesized hepcidin represents an interesting possibility, the physiological significance of such a regulation would depend on the ratio of cardiomyocyte-derived hepcidin synthesis versus systemic hepcidin synthesis. In this context, it has been reported that mouse heart hepcidin mRNA content is relatively high [18]. In the presented experiments, cardiac hepcidin expression was found to be rather variable, with typical PCR results showing a gap from 9 to 15 PCR cycles between the hepcidin cycle threshold and the reference gene cycle threshold. Thus, at the mRNA level, cardiac hepcidin expression in mice is apparently at least two orders of magnitude lower than liver hepcidin expression. It can therefore be speculated that systemic hepcidin synthesis is probably high enough to override the possible effect of cardiomyocyte-derived hepcidin, as documented by the observed increase in cardiac ferroportin content in erythropoietin-treated mice and rats.

In addition to erythropoietin administration, two additional models of low systemic hepcidin were examined—administration of a low-iron diet to rats and a mouse model of juvenile hemochromatosis. For the low-iron diet experiments, we chose young Wistar rats, since these animals reproducibly develop profound iron deficiency associated with dramatic hepcidin downregulation on low-iron diets [26]. While liver non-heme iron content decreased to about 10% of control levels in rats fed an iron-deficient diet, cardiac non-heme iron content decreased only approximately two-fold. Nevertheless, this decrease in cardiac non-heme iron content was apparently sufficient to decrease cardiac ferroportin expression, despite the marked decrease in systemic hepcidin. In accordance with a recently published rat study [32], we did not observe a significant effect of an iron-deficient diet on cardiac *Fpn* mRNA or *Hamp* mRNA content. These results suggest that cardiac ferroportin expression is primarily regulated by the IRP/IRE system, which seems to be capable of overriding the decrease in systemic hepcidin. Interestingly, the same apparent hierarchy in ferroportin protein regulation was observed in the liver.

Since heart failure is a prominent feature of untreated juvenile hemochromatosis [2], it was of interest to examine ferroportin protein content in mice with a targeted disruption of hemojuvelin (*Hjv*−/− mice), an animal model of Type 2A juvenile hemochromatosis. These mice are characterized by low hepatic hepcidin expression and severe iron overload. Cardiac non-heme iron content in *Hjv*−/− mice is reported to increase about five-fold [27]. Heart ferroportin protein content in these mice was significantly increased, which very probably results both from reduced IRP binding to the IRE sequence in *Fpn* mRNA as well as from the decrease in systemic hepcidin levels. In addition to this posttranscriptional regulation, *Hjv*−/− mice also displayed an increase in cardiac *Fpn* mRNA. Overall, these data indicate that cardiac FPN expression is regulated at both posttranscriptional and transcriptional levels.

During the past decade, it has been recognized that iron deficiency can have an adverse effect on heart failure outcomes, and current guidelines recommend the administration of iron carboxymaltose to iron-deficient heart failure patients [20]. At present, there is little information regarding the effect of iron carboxymaltose injection on liver, spleen, or heart ferroportin protein content. Despite the iron-induced increase in hepatic hepcidin mRNA, liver, and spleen ferroportin protein content displayed a marked increase in iron carboxymaltose-injected animals. In addition, treatment with iron carboxymaltose resulted in an increase in ferroportin protein content in the myocardium. Since the magnitude of the treatment-induced increases in ferroportin protein content appears to be proportional to the increases in tissue iron content, the results again suggest a dominant role of the IRP/IRE system in the regulation of cardiac ferroportin protein expression.

It is widely accepted that the hepcidin-ferroportin axis is the main determinant of tissue ferroportin protein content [1,13]. This is undoubtedly true in the intestine, where the regulation of enterocyte ferroportin by hepcidin plays a key role in the regulation of iron absorption, as evidenced by the pathophysiology of hereditary hemochromatosis. However, ferroportin protein regulation is much more complex [33], since ferroportin protein content is also regulated by the IRP/IRE system, as well as by iron-mediated changes in ferroportin transcription. In this respect, it should be noted that both IRE-containing and IRE-free ferroportin transcripts have been described [21], and the relative abundance of these two transcripts probably determines the overall response of ferroportin protein expression to cellular iron content. In cells with high expression of non-IRE transcripts, such as the enterocyte, the regulation of ferroportin by hepcidin represents the dominant pathway, which allows the precise regulation of dietary iron intake according to body iron status. In other tissues, such as the myocardium, the intracellular iron content might be an important regulator, as demonstrated by experiments in mice with cardiomyocyte-specific deletion of IRP1 and IRP2 proteins [34]. Overall, it is evident that the complex control of cellular iron export represents the major determining factor of iron homeostasis and that systemic hepcidin is an important but not the sole determinant in this process.

In conclusion, the study indicates that heart ferroportin protein expression is primarily regulated by heart iron content and secondarily by systemic hepcidin. Interestingly, the dominant effect of ferroportin protein regulation by intracellular iron is also observed in the liver and spleen. These data suggest that whereas the classical hepcidin/ferroportin axis plays a key role in the intestine, the expression of ferroportin protein in the heart, liver, and spleen is regulated predominantly by cellular iron status.

## 4. Materials and Methods

### 4.1. Animals and Treatment

All experiments were approved by the Ethics Committee of Czech Ministry of Education (protocol MSMT-11192/2020-2, dated 28 April 2020. Adult male C57BL/6J mice aged two to three months were purchased from commercial sources. Erythropoietin (NeoRecormon Roche) was administered at 50 U/mouse once daily for four consecutive days by intraperitoneal injection; animals were sacrificed 24 h after last treatment. Iron-carboxymaltose (Vifor, Paris, France) was administered at 300 mg Fe/kg by single intraperitoneal injection animals were sacrificed ten days after treatment.

In rat experiments, erythropoietin was administered at 600 U/rat for four days to outbred female Wistar rats (200–225 g) purchased from commercial sources; animals were sacrificed 24 h after last treatment.

For iron deficiency experiments, outbred young female Wistar rats with bodyweight 80–100 g were placed on an iron-deficient diet (C-1038, iron content 5 ppm, Altromin, Lage, Germany) for eight weeks; terminal body weight was 210–240 g. Control rats were fed a standard rodent diet containing approximately 200 ppm of iron.

*Hjv*−/− mice on mixed background [27] were a generous gift by Professor Silvia Arber, Basel, Switzerland. Male mice were sacrificed at 8 months of age.

### 4.2. Ferroportin Protein Determination

For the determination of ferroportin protein by immunoblots, we used the crude membrane fractions prepared by ultracentrifugation, as described by Canonne-Hergaux et al. [25,35]. Use of membranes instead of whole-cell lysates decreased the amount of non-specific bands and increased the intensity of ferroportin-specific bands. Tissues (about 150 mg) were homogenized in 10 volumes of buffered (pH 7.4) 0.25 M sucrose containing protease inhibitors and centrifuged for 15 min at 8000× *g*. Supernatants were transferred to ultracentrifuge tubes and centrifuged for one hour at 105,000× *g*. The resulting pellets were washed once by homogenization in 0.25 M sucrose buffer, recentrifuged at 105,000× *g*, and resuspended in 60 µL of 2% sodium dodecyl sulfate-containing 25 mM of ammonium bicarbonate.

In total, from 15 to 45 µg of protein was separated on 8% polyacrylamide gel. Samples were mixed with mercaptoethanol-containing loading buffer (Bio-Rad sro., Prague, Czech Republic) and heated for 5 min at 50 °C. Following electrophoresis, proteins were transferred to a PVDF membrane using Invitrogen XCell SureLock system (25 V, 120 min). The membrane was blocked for one hour in 5% skimmed milk. Ferroportin was detected using MTP11-A antibody from Alpha Diagnostics at 3:1000 dilution in skimmed milk, incubation with the primary antibody was for 48 h at 4 °C. Secondary antibody was Jackson Immunoresearch 711-036-152 at 1:40,000 dilution in skimmed milk, incubation was for two hours. Images were obtained on Biorad Chemidoc System (a total of 24 images at 5-min intervals at 4X binning setting) following signal visualization by Pierce ECL Western Blotting substrate (Thermo Fisher Scientific, Waltham, MA, USA). For loading control, N-Cadherin (#4061, Cell Signaling Technology Danvers, MA, USA) or GAPDH (G9545, Merck, Darmstadt, Germany antibodies were used. The membrane protein N-Cadherin was the preferred loading control; however, GAPDH was used in iron-overload experiments, since membrane N-Cadherin protein content is decreased by iron overload [36], and in spleen samples, where N-Cadherin was not detected.

To confirm that the bands used for ferroportin determination indeed represent ferroportin protein, a sample heated for 10 min to 85 °C was run on the same blot. It has been repeatedly demonstrated that determination of ferroportin requires only mild heating [21,25], and that boiling of the samples, as used in standard immunoblot protocols, markedly decreases the ferroportin protein content [37,38]. Therefore, the disappearance of a detected band following heating was regarded as a confirmation of that band specificity for ferroportin.

### 4.3. Real-Time PCR

Heart RNA was prepared from RNALater-stored samples using Trizol reagent (Thermo Fisher Scientific, Waltham, MA, USA), glycogen was added according to instructions to visualize the RNA pellet. Liver and spleen RNA was prepared using RNEasy Plus Mini Kit from Qiagen (Düsseldorf, Germany). Homogenization of all samples was performed using Roche Green Bead tubes, using 1 mL of the respective reagent (Trizol or RNEasy Plus buffer plus mercaptoethanol). Samples were reverse-transcribed using RevertAid kit (Thermo Fisher Scientific, Waltham, MA, USA). Real-time PCR was performed on Biorad IQ5 instrument using Biorad IQ SYBR Green mix; primer sequences are given in Appendix A. ΔCT values were calculated by subtracting the CT of the gene of interest from the CT of the reference gene; lower ΔCT in graphs means lower abundance of the target cDNA.

### 4.4. Iron Determinations

Tissue non-heme iron was determined according to [39]. Samples of approximately 50 mg were boiled in ten volumes of 10% trichloroacetic acid for 20 h at 65 °C; color was developed by bathophenantroline (Merck, Darmstadt, Germany and thioglycolic acid reagent. The use of non-heme iron determination diminished the risk of hemoglobin contamination in heart samples and excluded the contribution of myoglobin iron.

### 4.5. Statistical Analysis

Results were analyzed using two-tailed Students *t*-test with unequal variances; *p* ˂ 0.05 was regarded as significant.

## Figures and Tables

**Figure 1 ijms-23-05899-f001:**
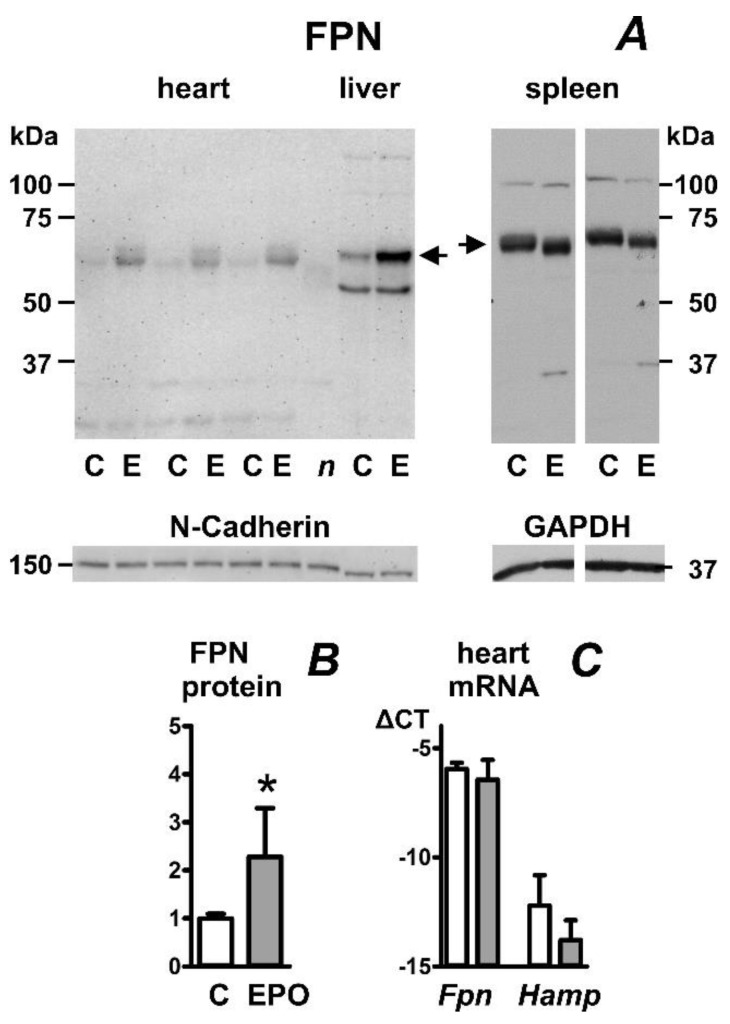
Effect of erythropoietin administration on ferroportin protein (FPN) content in mouse heart, liver and spleen. (**A**) Immunoblot of heart, liver, and spleen membrane protein fraction; (**B**) Quantification of heart immunoblot, n = 4; (**C**) Real-time PCR quantification of heart *Fpn* and *Hamp* mRNA expressed as ΔCT relative to *Actb* mRNA, n = 4. EPO was administered daily for four days at 50 U/mouse to male C57BL/6J mice; mice were sacrificed 24 h after last administration. C, samples from control mice; E, samples from EPO-treated mice. Samples were heated to 50 °C prior to loading; *n* (negative control) represents a sample from EPO-treated mouse heart, which was heated to 85 °C. N-Cadherin or GAPDH is used as loading control, arrows indicate the ferroportin-specific bands. Asterisk denotes statistical significance (*p* < 0.05).

**Figure 2 ijms-23-05899-f002:**
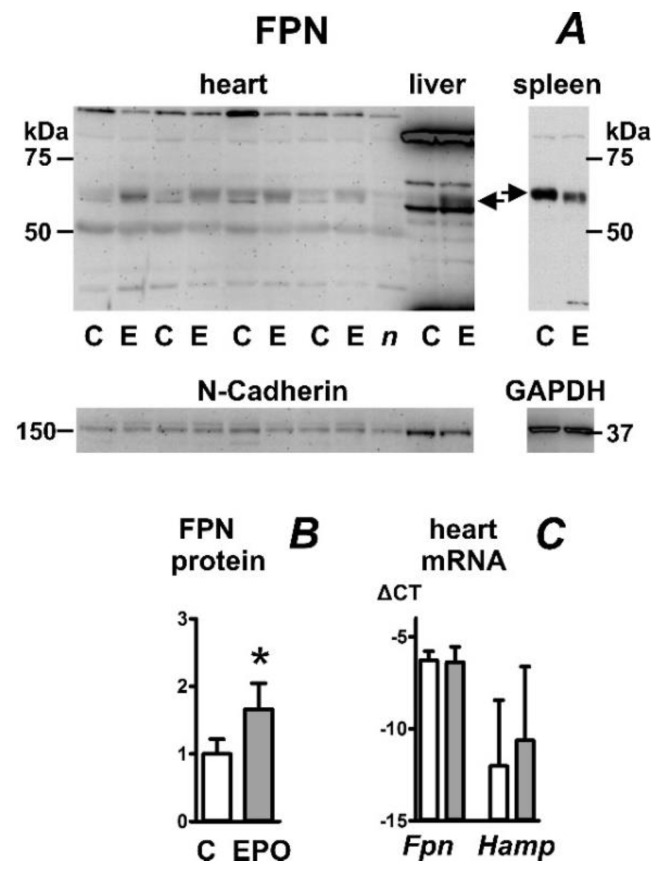
Effect of erythropoietin administration on ferroportin protein (FPN) content in rat heart, liver, and spleen. (**A**) Immunoblot of heart, liver, and spleen membrane protein fraction; (**B**) Quantification of heart immunoblot results, n = 4; (**C**) Real-time PCR quantification of heart *Fpn* and *Hamp* mRNA expressed as ΔCT relative to *Actb* mRNA, n = 4. EPO was administered to female Wistar rats daily for four days at 600 U/rat, animals were sacrificed 24 h after last administration. In total, 40 µg of membrane protein was loaded per well. C, samples from control rats; E, samples from EPO-treated rats. Samples were heated to 50 °C prior to loading; *n* (negative control) represents a heart sample from EPO-treated rat, which was heated to 85 °C. N-Cadherin or GAPDH is used as loading control, arrows indicate the ferroportin-specific bands. Asterisk denotes statistical significance (*p* < 0.05).

**Figure 3 ijms-23-05899-f003:**
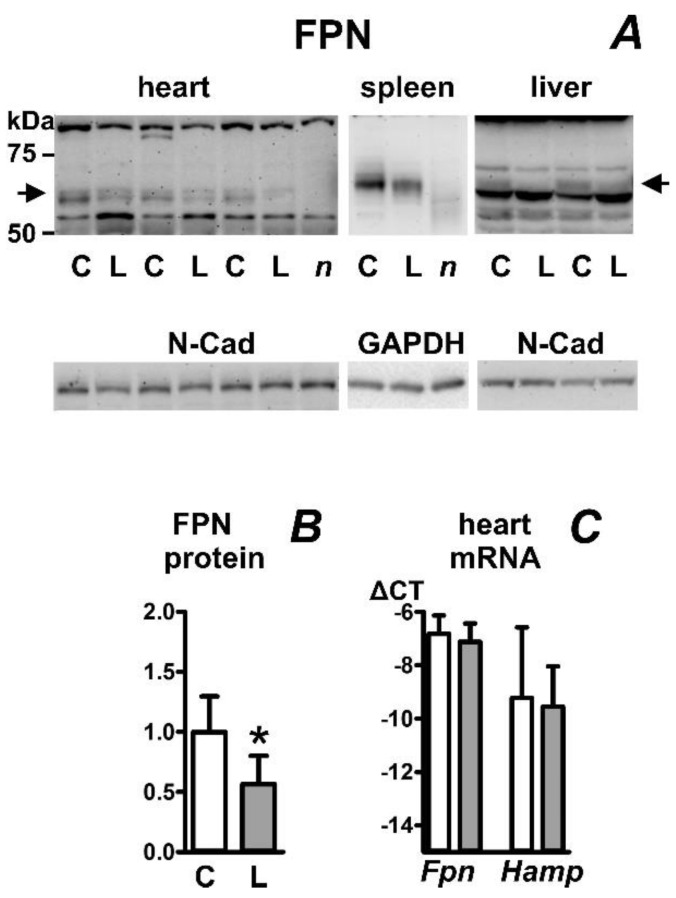
Effect of iron deficiency on ferroportin protein (FPN) content in rat heart, spleen, and liver. (**A**) Immunoblot of heart, spleen, and liver membrane protein fraction; (**B**) Quantification of heart FPN immunoblot results, n = 4; (**C**) Real-time PCR quantification of heart *Fpn* and *Hamp* mRNA expressed as ΔCT relative to *Actb* mRNA, n = 3. Young outbred female Wistar rats were placed on an iron-deficient diet for 8 weeks. Protein loading is 45 µg/well for heart, 15 µg/well for spleen and 30 µg/well for liver. C, samples from rats on control diet; L, samples from rats on iron-deficient diet. Samples were heated to 50 °C prior to loading; *n* (negative control) represents heart and spleen samples from control rat, which were heated to 85 °C. N-Cadherin (N-Cad, 150 kDa) or GAPDH (37 kDa) are used as loading controls; arrows indicate the ferroportin-specific bands. Asterisk denotes statistical significance (*p* < 0.05).

**Figure 4 ijms-23-05899-f004:**
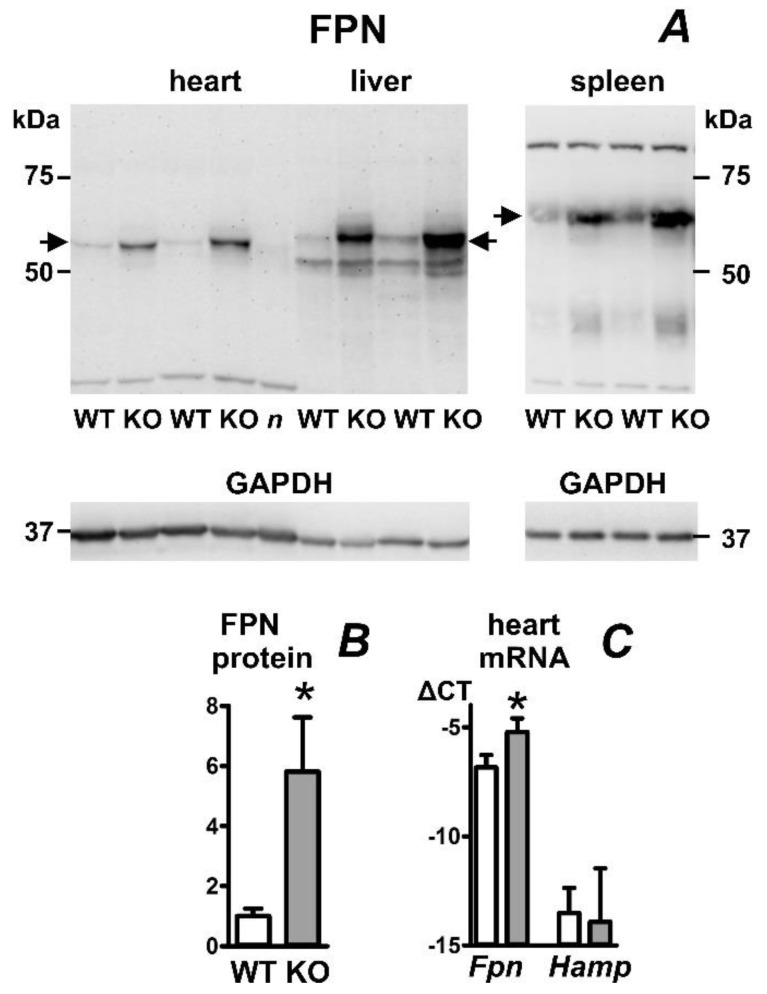
Effect of hemojuvelin gene disruption on ferroportin protein (FPN) content in mouse heart, liver, and spleen. (**A**) Immunoblot of heart, liver, and spleen membrane protein fraction; (**B**) Quantification of heart FPN immunoblot results, n = 4; (**C**) Real-time PCR quantification of heart *Fpn* and *Hamp* mRNA expressed as ΔCT relative to *Actb* mRNA, n = 4. Male *Hjv*+/+ and *Hjv*−/− mice were sacrificed at 8 months. In total, 40 µg of protein was loaded per well. WT, samples from *Hjv*+/+ mice; KO, samples from *Hjv*−/− mice. Samples were heated to 50 °C prior to loading; *n* (negative control) represents heart sample from *Hjv*−/− mouse, which was heated to 85 °C. GAPDH is used as loading control, arrows indicate the ferroportin-specific bands. Asterisk denotes statistical significance (*p* < 0.05).

**Figure 5 ijms-23-05899-f005:**
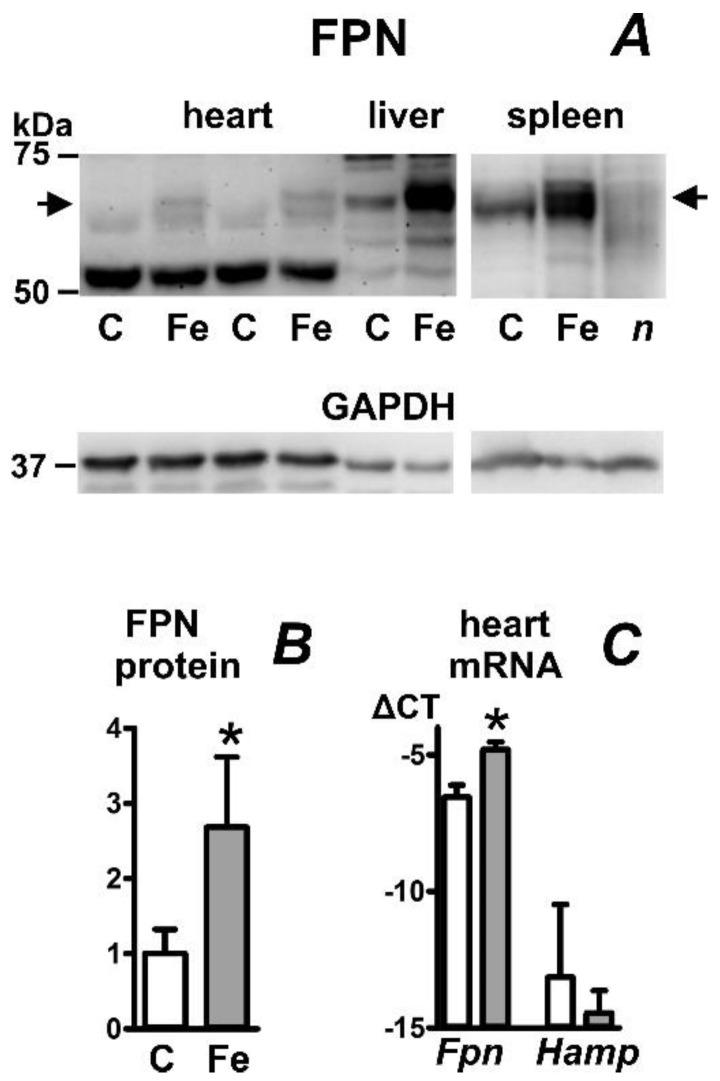
Effect of iron carboxymaltose injection on ferroportin protein (FPN) content in mouse heart, liver, and spleen. (**A**) Immunoblot of heart, liver, and spleen membrane protein fraction; (**B**) Quantification of heart FPN immunoblot results, n = 5; (**C**) Real-time PCR quantification of heart *Fpn* and *Hamp* mRNA expressed as ΔCT relative to *Actb* mRNA, n = 4. Male C57BL/6J mice were injected with a single dose of Ferinject (300 mg Fe/mouse) ten days prior to sacrifice. Protein loading was 40 µg per well for heart and liver samples and 15 µg per well for spleen samples. C, samples from control mice; Fe, samples from iron-injected mice. Samples were heated to 50 °C prior to loading; *n* (negative control) represents a spleen sample, which was heated to 85 °C. GAPDH is used as loading control, arrows indicate the ferroportin-specific bands. Asterisk denotes statistical significance (*p* < 0.05).

**Table 1 ijms-23-05899-t001:** Iron content and liver hepcidin expression in the experimental groups used.

Group	Heart Iron (µg/g)	Liver Iron (µg/g)	Spleen Iron (µg/g)	Liver *Hamp* (ΔCT)
Mouse, control	56.8 ± 5.6	79.5 ± 17.6	293.4 ± 55.9	−0.22 ± 0.62
Mouse, EPO	51.6 ± 0.4	54.6 ± 5.6	133.7 ± 50.1 *	−5.42 ± 2.90 *
Rat, control	42.7 ± 0.8	243.8 ± 21.9	729.7 ± 13.8	1.14 ± 0.35
Rat, EPO	47.1 ± 3.0	213.2 ± 74.9	195.0 ± 27.6 *	−6.45 ± 2.58 *
Rat, control diet	42.3 ± 5.4	363.3 ± 103.2	855.3 ± 105.8	0.83 ± 0.57
Rat, low Fe diet	24.2 ± 4.0 *	28.7 ± 4.1 *	31.3 ± 2.6 *	−14.49 ± 1.16
Mouse, *Hjv*+/+	58.5 ± 13.1	96.7 ± 33.0	1032.3 ± 423.0	0.54 ± 0.69
Mouse, *Hjv*−/−	463 ± 8.6 *	1952.1 ± 369.0 *	402.3 ± 126.4	−8.10 ± 2.30 *
Mouse, control	50.5 ± 1.5	73.5 ± 2.5	498.0 ± 124.0	0.49 ± 1.11
Mouse, Fe injection	316.1 ± 19.9 *	3020.5 ± 189.5 *	8249.3 ± 541.3 *	4.14 ± 0.54 *

Tissue non-heme iron is expressed per gram of wet weight, *Hamp* mRNA content is expressed as ΔCT relative to *Actb* mRNA. Asterisks denote statistical significance, n ≥ 3.

## Data Availability

Relevant data are included in the Figures, Table and Appendix A; additional data are available on request.

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
