# Peer review of "Heart Ferroportin Protein Content Is Regulated by Heart Iron Concentration and Systemic Hepcidin Expression"

_ijms, 2022, doi:10.3390/ijms23115899_

Round 1

Reviewer 1 Report

Hepcidin is a very interesting molecule in iron metabolism/homeostasis in various cells and tissues. The investigators sought to answer which hepcidin regulated cardiomyocyte ferroportin.

The manuscript is well structured and the topic has been well explored, but the description of the results needs to be clarified with the suggestions below.

Methods - for a more robust study, the number of animals used in the work and the number of replicates in the determinations must be indicated

Line 106 - need to review text. Tab.1 shows iron levels and non-expression of hepcidin. Confirm caption.

Line 115 – Figure S1 - Fig S1 must be inserted

Fig 1 Indicate the size of the N-Cadherin fragment

Line 117 - Figure 1. Effect of erythropoietin administration on mouse heart and liver ferroportin protein content.

I suggest - Figure 1. Effect of erythropoietin administration on mouse heart and liver ferroportin protein content. (A) Immunoblot of membrane protein fraction; (B) Quantification of heart immunoblot; (C) Real-time PCR quantification of heart Fpn and Hamp mRNA expressed as Δ CT relative to Actb. The caption must signal * for example p <0.05

line 141 – Figure 2. I suggest correction similar to line 117 (Fig 1)

Review legend for figures 3, 4, 5, similar to fig 1

Line 214 and 216 – Fig S2???

Author Response

We thank the Reviewer for the comments.

We made the following changes:

The numbers of animals in experiments were added where lacking.

Line 106: Caption to Table 1 was changed to include hepcidin expression.

Both Supplemental Figures were attached to the Marked Changes manuscript version  for easier access.

The size of N-Cadherin was added to Figure 1

All Figure legends were changed as requested.

Reviewer 2 Report

The publication by Berezovsky et al. submitted for review deals with an extremely important issue related to examine the response of heart ferroportin protein to changes in systemic hepcidin in experimental animals, and to compare heart ferroportin regulation with ferroportin regulation in the the liver and spleen. This is undoubtedly an important issue of great importance for clinicians (especially cardiologists), both in pathological and physiological states. This publication tries to some extent to revise the hypothesis that regarding the regulation of cardiomyocyte ferroportin instead of being regulated by systemic hepcidin, (as is the case for enterocyte ferroportin), cardiomyocyte ferroportin responds primarily to cardiomyocyte-produced hepcidin.

Five experiments in animal models were carried out to show how FPN levels change at different levels of systemic iron, which is a major determinant of hepcidin expressed in the liver. The results obtained suggests that, in contrast to ferroportin protein regulation in the duodenum, heart ferroportin protein content is determined primarily by heart iron concentration, whereas the regulation by hepcidin plays a secondary role.

Such conclusions are legitimate, however, a few shortcomings, which I list below, require clarification:

Significant shortcomings:

  • + IRE and -IRE FPN expression analyses were not performed.
  • Hepcidin peptide analysis required. Hepcidin is regulatated not only at transcriptional levels.
  • The binding of IRP proteins to IRE sequences has not been tested (EMSA). This would answer the authors' conjecture about the role of iron (LIP) in FPN regulation.
  • Please add splenic FPN protein levels in all experiments.

Questions:

  • What is the level of IRP1 between the heart and the liver. There are such literature data.
  • What is the level of FPN or iron in IRP1 KO mice in heart? (Bruno Galy papers maybe?
  • Could you explain different iron content between liver and spleen in Juvenile HH mouse model?

Minor shortcomings:

  • While I can agree on the differences in FPN mRNA between organs, the differences in Hamp mRNA expression are too small, even if it is only deltaCt. In In Lakhal-Littleton et al., (2016) it was 30x more Hamp mRNA in liver than in heart. Why are the expression differences not shown as fold change to control mice / rats?
  • When the complicated regulation of FPN expression is discussed, it is worth citing the immortal article by Drakesmith H, Nemeth E, Ganz T. Ironing out Ferroportin. Cell Metab. 2015 Nov 3; 22 (5): 777-87. doi: 10.1016 / j.cmet.2015.09.006. Epub 2015 Oct 1. PMID: 26437604; PMCID: PMC4635047.
  • Under reference number 20, I found no mention of FPN levels in anemic patients with acute heart failure.
  • In the introduction, it is worth emphasizing the fact that there are various forms of transcripts with or without IRE sequences. Regarding not only FPN and the heart.

Interestingly, the authors come to conclusions completely different from what Lakhal-Littleton in her two papers shows. Of course, the animal models she uses are artificial and do not reflect physiological phenomena. However, one must agree that knockout models are widely used. The approach used in this work is more physiological. I am also convinced that the local hepcidin in the heart may be important, but under specific conditions.

Considering the above, I believe that the current version of the manuscript meets the requirements of the IJMS journal and can be published prior taking into account proposed  changes. I believe that the conclusions reached by the authors are PARTIALLY justified in the presented research results, but could be far-reaching especially after taking into account my comments on the IRP / IRE system and hepcidin peptide analysis. I believe that the explanation of the role of local hepcidins in the regulation of iron metabolism in tissues in cooption with systemic hepcidin is another challenge that we face together in IRON MAN commonalty.

Best regards.

Author Response

We thank the Reviewer for the comments.

Regarding the Significant shortcomings:

We acknowledge that we did not perform the analysis of the IRE and non-IRE containing transcripts. The relative abundance of these transcripts in the liver, spleen and heart has been reported in Reference 21; the data indicate that there is no marked difference between these tissues.

We do not have the methodology for hepcidin determination in plasma. We think that it is safe to assume that the changes hepcidin peptide will correspond to the observed changes in liver Hamp mRNA, i.e. a decrease in EPO-treated groups, in rats on low-iron diet, and in Hjv-/- mice; Ferinject-treated mice will almost certainly display an increase in circulating hepcidin.  

At present, we do not have the methodology for the testing of IRP/IRE interactions.

We acknowledge that the additional information requested by the Reviewer would certainly significantly improve the manuscript; however, such changes would be hard to implement in the time frame of minor revision.  

Blots including splenic ferroportin have been added to Figures 1,2 and 4.

Regarding Questions:

1) We found no data reporting direct comparison of IRP1 expression in the liver and heart. Corna et al. (Corna G, Galy B, Hentze MW, Cairo G. IRP1-independent alterations of cardiac iron metabolism in doxorubicin-treated mice. J Mol Med (Berl). 2006 Jul;84(7):551-60. doi: 10.1007/s00109-006-0068-y) reports unchanged heart transferrin receptor and ferritin content in IRP1-deficient mice. However, combined deletion of both IRP1 and IRP2 increases ferroportin (see paragraph 2) below), which indicates that the IRP/IRE system functions in the heart.    

2) Ferroportin protein is reportedly increased in mice with cardiomyocyte-specific combined deletion of IRP1/IRP2, as reported by Haddad S, Wang Y, Galy B, et al. (Iron-regulatory proteins secure iron availability in cardiomyocytes to prevent heart failure. Eur Heart J. 2017;38(5):362-372. doi:10.1093/eurheartj/ehw333). This Reference (34) was added to the manuscript.

3) The possible reason for the difference between spleen iron concentration in Hjv-/- and Ferinject-injected mice was included (lines 217-220).  Mouse hemochromatosis models display low hepcidin, which certainly contributes to decreased iron retention in the spleen; however, the main reason for the increased splenic iron content in Ferinject-injected mice is probably the grossly increased availability of various forms of iron.  

Regarding Minor Shortcomings:

When comparing real-time PCR results, we prefer the actual Δ CT values to relative comparison with the controls.  Δ CT provides more information – it not only allows the comparison between control and treated groups, but it also enables evaluation of the basal expression of the gene.

The paper by Drakesmith et al has been included as Reference 33

The sentence regarding Reference 20 (lines 77-80) has been re-formulated.

The existence of IRE and non-IRE isoforms is not mentioned in the Introduction, but it is discussed in the Discussion.  

Overall, we would like to state that we certainly acknowledge the fact that much more could have been done regarding IRE/IRP binding. Our manuscript focuses primarily on FPN protein determination, since, in our view, many papers report „ferroportin“ protein bands without taking adequate measures to ensure that the reported bands indeed represent ferroportin. Investigation of IRE/IRP binding would indeed be a logical continuation of our manuscript.